# Phase II Study of Atezolizumab and Bevacizumab Combination Therapy for Patients with Advanced Hepatocellular Carcinoma Previously Treated with Lenvatinib

**DOI:** 10.3390/cancers17020278

**Published:** 2025-01-16

**Authors:** Takeshi Terashima, Hidenori Kido, Noboru Takata, Tomoyuki Hayashi, Akihiro Seki, Hidetoshi Nakagawa, Kouki Nio, Tadashi Toyama, Noriho Iida, Shinya Yamada, Tetsuro Shimakami, Hajime Takatori, Kuniaki Arai, Tatsuya Yamashita, Eishiro Mizukoshi, Taro Yamashita

**Affiliations:** 1Department of Gastroenterology, Kanazawa University Hospital, Kanazawa 920-8641, Ishikawa, Japan; 2Department of Nephrology, Faculty of Medical Sciences, University of Fukui, Fukui 910-1104, Fukui, Japan

**Keywords:** hepatocellular carcinoma, atezolizumab, bevacizumab, second-line treatment

## Abstract

In this phase II trial, patients with advanced hepatocellular carcinoma (HCC) previously treated with lenvatinib were enrolled to receive atezolizumab and bevacizumab every 3 weeks. The primary endpoint was progression-free survival. A total of 26 eligible patients were enrolled. The median progression-free survival from the start of treatment was 9.70 [90% confidence interval, 5.10–14.24] months, with the lower limit above the predefined threshold. The median overall survival was 17.23 [90% confidence interval, 13.18–27.85] months and the objective response rate was 34.6%. Sixteen patients (61.5%) received subsequent therapies. Severe adverse events, adverse events leading to treatment delays, and adverse events leading to treatment discontinuation occurred in eight (30.8%), fourteen (53.8%), and five (19.2%) patients, respectively, and no treatment-related death occurred. This combination therapy is suggested to be effective and safely administered for patients with advanced HCC previously treated with lenvatinib.

## 1. Introduction

Hepatocellular carcinoma (HCC) is the fifth leading cause of cancer-related death in Japan (https://www.mhlw.go.jp/toukei/saikin/hw/jinkou/kakutei23/index.html, accessed on 30 November 2024) and the third leading cause of cancer-related death worldwide [1]. Successful development of agents that suppress or eradicate the hepatitis virus causing carcinogenesis can lead to a reduction in HCC risk in certain patients [2,3], and the identification of high-risk patients using novel biomarkers allows for the early detection of HCC at the curative stage during routine surveillance [4,5]. However, frequent recurrence even after curative treatment eventually becomes difficult to treat using locoregional therapy [6].

Several systemic therapies have been successfully developed in the last decade for advanced HCC patients. First, sorafenib significantly improves overall survival (OS) compared to a placebo in such patients [7]. Although several drugs failed to verify their efficacies for a period afterward, three clinical trials found that regorafenib [8], cabozantinib [9], and ramucirumab [10] provided survival benefits compared to a placebo as second- or later-line treatment in patients with advanced HCC who had previously received sorafenib. In addition, lenvatinib showed comparable efficacy to sorafenib as a first-line treatment in the REFLECT study [11], and its effectiveness has been also demonstrated in a systematic review and meta-analysis [12]. Most similar clinical studies showed that lenvatinib was statistically superior to sorafenib in direct antitumor efficacy indicators such as objective response rate (ORR) and progression-free survival (PFS), but not in OS.

One possible factor that could explain this discrepancy is that there may be differences in the effectiveness of post-treatment. For advanced HCC patients, it has been shown that second- and later-line treatments as well as first-line treatment play a significant role in patient outcomes [13,14]. Post hoc analysis of the REFLECT study also suggested that subsequent therapy after lenvatinib contributed to prolonging patients’ prognosis [15]; however, the five drugs mentioned above that were available to patients were all classified as molecularly targeted drugs, and no studies to date have demonstrated that any treatment improves the prognosis of patients previously treated with lenvatinib.

IMbrave150, a phase III trial, demonstrated that combination therapy using atezolizumab and bevacizumab has demonstrated effectiveness compared to sorafenib for patients with advanced HCC [16], and the combination has been established as the standard first-line treatment [17,18]. However, the trial only included patients with advanced HCC who did not receive systemic therapy; therefore, its efficacy and safety have not been fully ensured for patients undergoing prior systemic therapy.

Immune checkpoint inhibitors, mainly anti-programmed death ligand-1 antibodies such as atezolizumab or anti-programmed death 1 antibodies, have a different mode of action from conventional cytotoxic agents and molecular targeted drugs and have brought about revolutionary advances in various types of cancer. Their efficacy has been demonstrated even after disease progression on tyrosine kinase inhibitors in other carcinomas [19,20]. Under these circumstances, combination immunotherapies using atezolizumab and bevacizumab are considered another treatment option for such patients in the clinical setting.

Therefore, this phase II trial was conducted to evaluate the efficacy and safety of atezolizumab and bevacizumab combination therapy for patients with advanced HCC previously treated with lenvatinib. The results of this trial will provide significant information when considering the treatment sequence for advanced HCC.

## 2. Materials and Methods

### 2.1. Patients

This open-labeled, single-arm, prospective phase II trial (jRCT1041200068) included patients clinically diagnosed with HCC who were unsuitable for local therapy, had previously received systemic therapy, had preserved liver function, were ≥20 years old, had an Eastern Cooperative Oncology Group performance status (ECOG PS) of 0, 1, or 2, had preserved major organ function, and had provided written informed consent. Conversely, patients who had refractory ascites or pleural effusion, esophageal or gastric varices with a high risk of bleeding, double cancer, metastases to the central nervous system, severe complications, mental disorder, were pregnant or lactating, had childbearing potential, or were regarded by the investigator as unfit were excluded.

### 2.2. Protocol Treatment and Assessment of Safety and Efficacy

Treatment consisted of a dose of 1,200 mg of atezolizumab plus 15 mg/kg of bevacizumab every 3 weeks, which was based on the treatment schedule used in the IMbrave150 trial [16]. Dose reduction was not permitted for either agent. Treatment was postponed if needed, including when grade 3 or 4 toxicity was observed, until the predefined criteria were met, such as resolution to grade 1 or less, and continued until radiological tumor progression, unacceptable toxicity, patient refusal, or death.

HCC was diagnosed according to the radiological findings of hyperattenuation in the arterial phase and hypoattenuation in the late phase, which were characterized using triple-phase contrast-enhanced computed tomography (CT) or gadolinium ethoxybenzyl diethylenetriamine pentaacetic acid-enhanced magnetic resonance imaging (MRI) following the guidelines of the American Association for the Study of Liver Disease [21]. All patients underwent CT from the chest to the pelvis to assess the extent of HCC. For the efficacy assessment, the patients received dynamic CT or dynamic MRI every 6 weeks after treatment started. The antitumor effects were assessed in accordance with the Response Evaluation Criteria in Solid Tumors (RECIST) ver. 1.1. All patients were provided with comprehensive information about the risks and benefits of these treatments, including the known safety and efficacy of the treatment, and they provided written informed consent.

### 2.3. Data Collection

Demographic, clinical, and laboratory data were collected prospectively. The adverse events were evaluated according to the Common Terminology Criteria for Adverse Events ver. 5.0. The Institutional Review Board of Kanazawa University approved this study, which was conducted in accordance with the Declaration of Helsinki.

### 2.4. Statistical Analysis

The primary endpoint was PFS, and the key secondary endpoint was OS. The other secondary endpoints included the 6-month PFS rate, 1-year OS rate, ORR, disease control rate (DCR), subsequent therapy, and frequency of adverse events. The threshold and expected PFS were 3 and 6.8 months, respectively. During the enrollment period of 2 years and considering a one-sided significance level of 0.05 and a statistical power of 80%, the minimum required sample size was 26 patients. Patients previously treated with lenvatinib were set as the primary analysis population.

PFS was defined as the duration from treatment initiation until the date of radiological progression, death, or the last day of follow-up. OS was defined as the period from treatment initiation until death. To compare PFS and OS between subgroups, the cumulative survival proportions were calculated using the Kaplan–Meier method, and any differences were evaluated using Cox’s proportional hazards regression model. A *p* value less than 0.05 was considered statistically significant. ORR was defined as the sum of the complete response rate and the partial response rate according to RECIST ver. 1.1. All statistical analyses were performed using Stata 14.2 (Stata Corp, College Station, TX, USA).

## 3. Results

### 3.1. Patient Characteristics

The first patient was enrolled in November 2020, and 26 eligible patients were enrolled until September 2022. All patients received at least one cycle of protocol treatment and were included in the population for the safety and efficacy analysis set. The patient characteristics are summarized in Table 1. The median age was 73.5 years, and all of the enrolled patients were male. All four patients who tested positive for hepatitis B virus surface antigen had been treated with nucleos(t)ide analog, and all three patients who tested positive for hepatitis C virus antibodies and subsequently eradicated the virus achieved a sustained virological response.

Among the enrolled patients, nine (34.6%), six (23.1%), and fourteen (53.8%) patients were treated with resection, local ablation, and transarterial chemoembolization, respectively. Regarding prior systemic therapy, all patients received lenvatinib, and eight (30.8%) and six (23.1%) patients were also treated with sorafenib and regorafenib, respectively (Table 2).

### 3.2. Treatment

The data collection cutoff was 12 October 2023, which was 1 year after the last patient started the treatment, and the median follow-up was 16.3 (range, 1.0–33.7) months. The median treatment duration of atezolizumab and bevacizumab combination therapy was 7.6 (range, 0.5–24.9) months, and the median number of atezolizumab and bevacizumab treatments were 9 (range, 1–36) and 6.5 (range, 1–27), respectively. All patients experienced treatment failure due to tumor progression (n = 19), unacceptable adverse events (n = 5), and treatment refusal (n = 2).

Sixteen patients (61.5%) received one or more subsequent therapies after atezolizumab and bevacizumab combination therapy. The most frequently performed subsequent therapies were durvalumab and tremelimumab combination therapy (n = 5) and radiotherapy (n = 5), followed by ramucirumab (n = 4), sorafenib (n = 4), hepatic arterial infusion chemotherapy (n = 4), transarterial chemoembolization (n = 4), and radiofrequency ablation (n = 3).

### 3.3. PFS and OS

Figure 1 shows the Kaplan–Meier curve of the PFS. The median PFS from the start of treatment was 9.70 [90% confidence interval (CI), 5.10–14.24] months, and the lower limit of the 90% CI was above the predefined threshold. The 6-month and 1-year PFS rates were 60.0% and 36.0%, respectively. Subgroup analysis of PFS did not find significant differences in PFS between groups regarding patient’s background including age, ECOG PS, etiology of chronic liver disease, Child–Pugh classification, maximum tumor size, vascular invasion, extrahepatic lesion, serum α-fetoprotein level, prior systemic therapy such as treatment line of prior systemic therapy, reasons for discontinuation of prior lenvatinib therapy, response to lenvatinib, and treatment duration of lenvatinib (Table 3).

Among the enrolled patients, 17 (65%) had died because of tumor progression (n = 14), rupture of esophageal varices (n = 2), and liver failure (n = 1) at the data collection cutoff; these deaths were found to have no relationship to protocol treatment. Figure 2 shows the Kaplan–Meier curve of the OS. The median OS from the start of treatment was 17.23 (90% CI, 13.18–27.85) months. The 6-month and 1-year OS rates were 80.8% and 69.2%, respectively. The subgroup analysis of OS also found no significant differences in the OS between the groups regarding the patient’s background or prior systemic therapy (Table 4).

### 3.4. Radiological Response to Atezolizumab and Bevacizumab

The radiological responses to atezolizumab and bevacizumab combination therapy in individual patients are shown in Figure 3 and summarized in Table 5. The ORR and DCR were observed in 34.6% and 73.1% of the patients, respectively. In some of the patients who finally achieved a response, the tumor initially showed slight enlargement 6 weeks after treatment commenced and then began to shrink (Figure 4). In addition, once a response was achieved, the anti-tumor effect was maintained for a long period of time.

### 3.5. Adverse Events

The safety profiles of the atezolizumab and bevacizumab combination therapy are summarized in Table 6. Severe adverse events were observed in eight (30.8%) patients, including ascites in two, and generalized muscle weakness, cognitive disturbance due to hepatic encephalopathy, heart failure, pneumonitis, esophageal varices hemorrhage, and tumor hemorrhage in one. Of these, generalized muscle weakness and pneumonitis were determined to be immune-related adverse events caused by atezolizumab, and cognitive disturbance, heart failure, esophageal varices hemorrhage, and tumor hemorrhage were caused by bevacizumab. Outcomes for all severe adverse events were recovery or improvement, and no treatment-related deaths were observed. Adverse events leading to treatment delays occurred in 14 (53.8%) patients. The most prevalent adverse events leading to treatment delay were proteinuria (n = 5), encephalopathy (n = 2), ascites (n = 2), and liver injury (n = 2). Outcomes for all adverse events led to treatment delay, but ascites led to recovery or improvement. Adverse events leading to treatment discontinuation occurred in five (19.2%) patients, including immune-related neurological disorder (n = 1), worsening general condition (n = 1), spontaneous bacterial peritonitis (n = 1), ascites (n = 1), and pneumonitis (n = 1). With the exception of ascites, outcomes for all adverse events leading to treatment discontinuation were recovery or improvement, and no treatment-related death occurred.

## 4. Discussion

This study primarily aimed to evaluate the efficacy of this combination therapy. As a result of this study, the primary endpoint was achieved, and the efficacy results including a median PFS of 9.70 months, 6-month and 1-year PFS rates of 60.0% and 36.0%, and an ORR of 34.6% could indicate that atezolizumab and bevacizumab combination therapy is also effective in patients with advanced HCC previously treated with lenvatinib. In several previous observational studies, sorafenib, regorafenib, and ramucirumab were candidates for post-treatment with lenvatinib; however, their efficacy was very limited, and the median PFS of the patients was between 1.8 and 3.2 months [22,23]. Because the effectiveness of these agents is attributed to the suppression of the signaling that is also targeted by lenvatinib, their less favorable outcomes in patients who were previously treated with lenvatinib, when compared with their effectiveness in lenvatinib-naïve patients, can be explained by their similar molecular targeting profiles and pharmacologic activity. From this perspective, atezolizumab has a different mechanism of action compared with lenvatinib, which can explain its effectiveness in patients previously treated with lenvatinib that is comparable to that in patients who did not receive systemic therapy.

Some retrospective studies noted that the therapeutic effect of atezolizumab and bevacizumab combination therapy in unselected patients with advanced HCC treated with lenvatinib was inferior to that in patients not treated with lenvatinib [24,25,26]. In contrast, the efficacy results of the present study were comparable with those of the IMbrave150 trial and other studies describing the treatment efficacy of atezolizumab and bevacizumab combination therapy in clinical practice. Although the reasons for this favorable outcome were unclear, a possible factor is the selection of the enrolled patients in this study. Although the inclusion and exclusion criteria for this study are commonly used, as described in the methods section, patients enrolled in clinical trials were generally in good general condition. In addition, the relatively high response rate to previously administered lenvatinib and the 8.7 months of treatment duration may indicate that many of the study participants might have had less aggressive tumors. Conversely, half of the enrolled patients had Child–Pugh B at treatment initiation, and these patients have been reported to have an inferior response to atezolizumab and bevacizumab combination therapy compared to patients with Child–Pugh A [27]. We cannot reach a definite conclusion because the single-arm nature of the study requires caution in interpreting time-to-event results such as PFS. However, the ORR of 34.6% and DCR of 73.1% as well as the median OS of 17.23 months signified that atezolizumab and bevacizumab combination therapy is an important treatment option for selected patients with advanced HCC previously treated with lenvatinib.

In this study, the safety profile of atezolizumab and bevacizumab combination therapy, such as severe adverse events or adverse events requiring dose adjustment, was well known, and no novel signals were detected. They were categorized as adverse events caused by bevacizumab and immune-related adverse events caused by atezolizumab [28]. Pneumonitis and neurological disorder leading to generalized muscle weakness occurred as immune-related adverse events, and both of them can be resolved by the administration of high-dose corticosteroids. However, bevacizumab-related adverse events such as bleeding, heart failure, and portal hypertension including encephalopathy and ascites were worth noting. Recently, long-term administration of high-dose bevacizumab was reported to decrease liver functional reserve [28]. Bleeding or cardiovascular events are well-known bevacizumab-induced adverse events, and treatment outcomes were expected to be favorable when prophylactic procedures were performed for patients with high-risk esophagogastric varices, along with appropriate monitoring and care [29]. Of course, the safety of atezolizumab and bevacizumab combination therapy for patients previously treated with systemic therapy cannot be established based only on the results of this study.

This study has several limitations such as its non-comparative design including a small population from a single institution. However, to the best of our knowledge, this is the first study evaluating the safety and efficacy of atezolizumab and bevacizumab combination therapy for patients previously treated with systemic therapy based on the predefined hypothesis; thus, further investigations are needed to confirm the presented findings.

## 5. Conclusions

The results imply that atezolizumab and bevacizumab combination therapy is effective and can safely be administered to patients with advanced HCC previously treated with lenvatinib. However, it is necessary to confirm whether our findings can be implemented in clinical practice for patients.

## Figures and Tables

**Figure 1 cancers-17-00278-f001:**
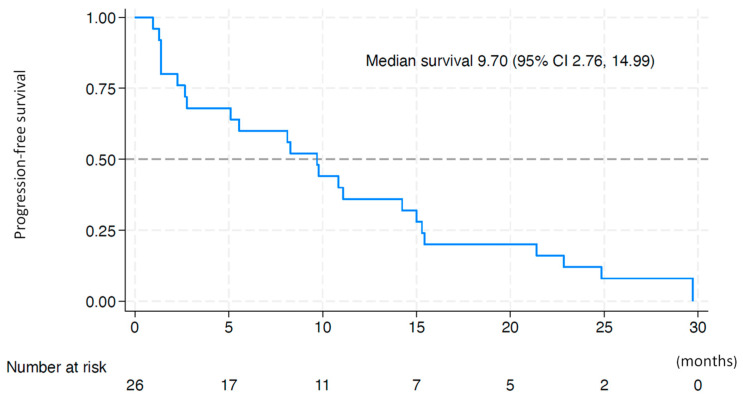
Kaplan–Meier plot of progression-free survival after treatment commenced. The median progression-free survival from the start of treatment was 9.70 (90% confidence interval, 5.10–14.24) months.

**Figure 2 cancers-17-00278-f002:**
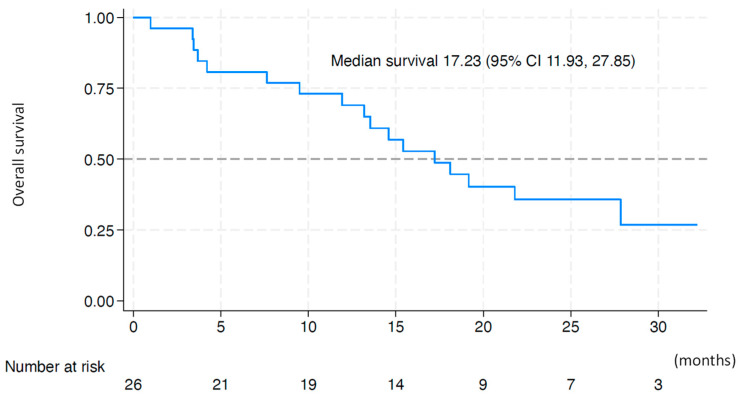
Kaplan–Meier plot of overall survival after treatment commenced. The median overall survival from the start of treatment was 17.23 (90% confidence interval, 13.18–27.85) months.

**Figure 3 cancers-17-00278-f003:**
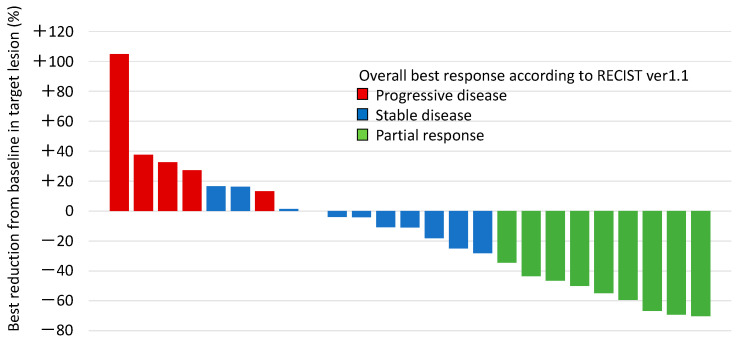
Waterfall plot showing best reduction in target lesion from baseline.

**Figure 4 cancers-17-00278-f004:**
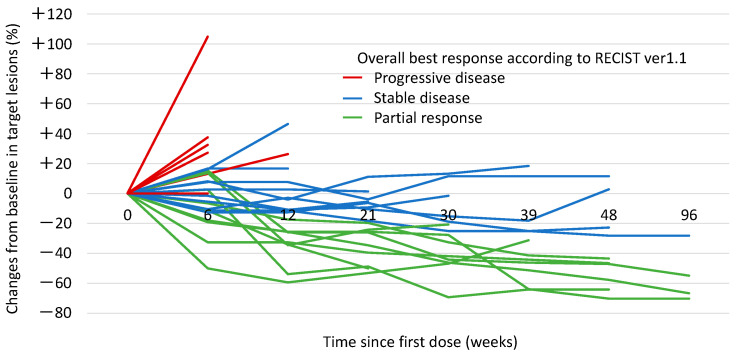
Spider plot showing changes in target lesions during treatment.

**Table 1 cancers-17-00278-t001:** Patient characteristics.

	Patients, n	(%)
Age, years		
Median	73.5	
Range	48–87	
Sex		
Male	26	(100)
Female	0	(0)
Eastern Cooperative Oncology Group performance status		
0	19	(73.1)
1	5	(19.2)
2	2	(7.7)
Etiology of chronic liver disease		
Hepatitis B virus surface antigen, positive	4	(15.4)
Hepatitis C virus antibody, positive	3	(11.5)
Child–Pugh score at treatment initiation		
Score 5	5	(19.2)
Score 6	8	(30.8)
Score 7	13	(50.0)
Maximum tumor size, mm		
Median	31	
Range	12–157	
Vascular invasion, presence	6	(23.1)
Extrahepatic lesion, presence	11	(42.3)
Serum α-fetoprotein level, ng/mL		
Median	46.5	
Range	4–222,669	

**Table 2 cancers-17-00278-t002:** Prior systemic therapy.

	Patients, n	(%)
Prior treatment	26	(100)
Resection	9	(34.6)
Local ablation	6	(23.1)
Transarterial chemoembolization	14	(53.8)
Systemic treatment	26	(100)
Prior systemic therapy		
Lenvatinib	26	(100)
Sorafenib	8	(30.8)
Regorafenib	6	(23.1)
Treatment line of prior systemic therapy		
1	13	(50.0)
2	5	(19.2)
≥3	8	(30.8)
Reasons for the discontinuation of prior lenvatinib therapy		
Tumor progression	19	(73.1)
Unacceptable adverse events	7	(26.9)
Response to prior lenvatinib therapy according to the RECIST		
Partial response	9	(34.6)
Stable disease	17	(65.4)
Treatment duration of prior lenvatinib therapy, months		
Median	8.7	
Range	0.4–19.3	

**Table 3 cancers-17-00278-t003:** Subgroup analysis of progression-free survival.

	Patients, n	Hazard Ratio (95% C.I.)	*p* Value *
Age, years old			0.66
≥74/<74	13/13	1.202 (0.529–2.733)	
Eastern Cooperative Oncology Group performance status			0.42
0/1 or 2	19/7	0.676 (0.263–1.735)	
Hepatitis B virus surface antigen			0.97
Positive/Negative	4/22	1.020 (0.343–3.028)	
Hepatitis C virus antibody			0.84
Positive/Negative	3/23	0.881 (0.259–3.004)	
Child–Pugh classification at the start of treatment			0.73
A/B	13/13	1.157 (0.508–2.638)	
Maximum tumor size, mm			0.65
≥31/<31	12/14	1.219 (0.524–2.833)	
Vascular invasion			0.58
Presence/Absence	6/20	0.747 (0.268–2.081)	
Extra-hepatic lesion			0.63
Presence/Absence	11/15	1.227 (0.531–2.838)	
Serum α-fetoprotein level, ng/mL			0.99
≥46.5/<46.5	13/13	0.995 (0.427–2.317)	
Treatment line of prior systemic therapy			0.48
≥2/1	13/13	1.363 (0.577–3.220)	
Reasons for discontinuation of prior lenvatinib therapy			0.27
Unacceptable adverse events/Tumor progression	7/19	0.566 (0.208–1.545)	
Response to prior lenvatinib therapy according to RECIST			0.60
Partial response/Stable disease	9/17	0.794 (0.333–1.895)	
Treatment duration of prior lenvatinib therapy, months			0.72
≥8.7/<8.7	13/13	0.858 (0.374–1.966)	

C.I., confidence interval; *: Log-rank test.

**Table 4 cancers-17-00278-t004:** Subgroup analysis of overall survival.

	Patients, n	Hazard Ratio (95% C.I.)	*p* Value *
Age, years old			0.89
≥74/<74	13/13	1.070 (0.412–2.778)	
Eastern Cooperative Oncology Group performance status			0.59
0/1 or 2	19/7	0.729 (0.234–2.273)	
Hepatitis B virus surface antigen			0.89
Positive/Negative	4/22	0.895 (0.204–3.932)	
Hepatitis C virus antibody			0.64
Positive/Negative	3/23	0.703 (0.160–3.085)	
Child–Pugh classification at the start of treatment			0.95
A/B	13/13	0.971 (0.373–2.529)	
Maximum tumor size, mm			0.24
≥31/<31	12/14	1.771 (0.676–4.644)	
Vascular invasion			0.79
Presence/Absence	6/20	0.845 (0.240–2.978)	
Extra-hepatic lesion			0.51
Presence/Absence	11/15	1.387 (0.523–3.674)	
Serum α-fetoprotein level, ng/mL			0.56
≥46.5/<46.5	13/13	1.329 (0.505–3.493)	
Treatment line of prior systemic therapy			0.33
≥2/1	13/13	0.616 (0.232–1.637)	
Reasons for discontinuation of prior lenvatinib therapy			0.69
Unacceptable adverse events/Tumor progression	7/19	1.234 (0.434–3.511)	
Response to prior lenvatinib therapy according to RECIST			0.75
Partial response/Stable disease	9/17	0.846 (0.297–2.408)	
Treatment duration of prior lenvatinib therapy, months			0.72
≥8.7/<8.7	13/13	0.858 (0.374–1.966)	

C.I., confidence interval; *: Log-rank test.

**Table 5 cancers-17-00278-t005:** Response to treatment according to RECIST ver1.1.

	Patients, n	(%)
Complete response	0	
Partial response	9	(34.6)
Stable disease	10	(38.5)
Progressive disease	6	(23.1)
Not evaluable	1	(3.8)
Objective response rate	34.6%	
Disease control rate	73.1%	

RECIST, Response Evaluation Criteria in Solid Tumors.

**Table 6 cancers-17-00278-t006:** Safety profiles of the atezolizumab and bevacizumab combination therapy.

	Patients, n	(%)
Severe AEs	8	(30.8)
Ascites	2	(7.7)
Generalized muscle weakness	1	(3.8)
Cognitive disturbance	1	(3.8)
Heart failure	1	(3.8)
Pneumonitis	1	(3.8)
Esophageal varices hemorrhage	1	(3.8)
Tumor hemorrhage	1	(3.8)
AEs leading to treatment delays	14	(53.8)
Proteinuria	5	(19.2)
Encephalopathy	2	(7.7)
Ascites	2	(7.7)
Liver injury	2	(7.7)
Cough	1	(3.8)
Blood bilirubin increased	1	(3.8)
Lung infection	1	(3.8)
Heart failure	1	(3.8)
Cognitive disturbance	1	(3.8)
Esophageal varices hemorrhage	1	(3.8)
Tumor hemorrhage	1	(3.8)
AEs leading to treatment discontinuation	5	(19.2)
Neurological disorder	1	(3.8)
Worsening general condition	1	(3.8)
Spontaneous bacterial peritonitis	1	(3.8)
Ascites	1	(3.8)
Pneumonitis	1	(3.8)

AE, adverse events.

## Data Availability

The data presented in this study are available in this article.

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
