# Peer review of "Phase II Study of Atezolizumab and Bevacizumab Combination Therapy for Patients with Advanced Hepatocellular Carcinoma Previously Treated with Lenvatinib"

_cancers, 2025, doi:10.3390/cancers17020278_

Round 1
Reviewer 1 Report
Comments and Suggestions for Authors
In a Phase II study, the authors analyzed the efficacy and safety of the combination therapy of Atezolizmab + Bevacizumab with Lenvatinib for the treatment of hepatocellular carcinoma, and reported that it was highly effective and could be a new option after Lenvatinib therapy. Since there are few papers reporting the efficacy of Atezolizumab + Bevacizumab after Lenvatinib, this is an important analysis and a useful report for specialists involved in the treatment of hepatocellular carcinoma.
Minor comments
1. Is there any difference in efficacy and safety between patients who started Atezo/Bev as 2nd line after Lenvatinib and those who started it as 3rd or 4th line? The number of cases is limited, so it may be difficult to analyze, but please report if possible.
2. Are you meaning that the proteinuria in the AEs causing the prolonged treatment in Table 4 was reversible with Atezo/Bev withdrawal?
Author Response
- Is there any difference in efficacy and safety between patients who started Atezo/Bev as 2nd line after Lenvatinib and those who started it as 3rd or 4th line? The number of cases is limited, so it may be difficult to analyze, but please report if possible.
Response
We appreciated your important indication regarding the differences of efficacy of atezolizumab plus bevacizumab according to treatment line. Following your comment, we have performed the subgroup analysis in PFS and OS according to patients’ background, and found the results showed that there was no efficacy difference between the patients received atezolizumab plus bevacizumab combination therapy as 2nd line and 3rd or later line. We added the subgroup analysis of PFS and OS to Table 3 and Table 4, respectively.
- Are you meaning that the proteinuria in the AEs causing the prolonged treatment in Table 4 was reversible with Atezo/Bev withdrawal?
Response
We appreciated your important question. As you assumed, all outcomes of the proteinuria were recovery or improvement. We have added the missing information on the outcomes of adverse events leading to treatment delays and treatment discontinuation as below.
“Outcomes for all adverse events leading to treatment delay but ascites were recovery or improvement.” and “Outcomes for all adverse events leading to treatment discontinuation but ascites were recovery or improvement and no treatment-related death occurred.” to the results section.
Reviewer 2 Report
Comments and Suggestions for Authors
Reviewer Comments
This manuscript presents a Phase II clinical study evaluating atezolizumab (anti-PD-L1) and bevacizumab (anti-VEGF) in patients with advanced-stage hepatocellular carcinoma (HCC) previously treated with lenvatinib. The study provides valuable insights into the efficacy of systemic therapies for HCC. However, the manuscript could be strengthened by addressing the following points:
- Post-Treatment Data Visualization
While the authors provide detailed information on patients' baseline characteristics, there is limited data presented on patients' outcomes during and after treatment. Specifically, only two Kaplan-Meier plots (Figures 1 and 2) are included. To better inform clinicians and researchers, the addition of more detailed post-treatment data is recommended. For example, incorporating a waterfall plot (e.g., Figure 2 from Yau et al.) and a spider plot (e.g., Figure 3 from Yau et al.) would allow for a more comprehensive visualization of individual patient responses and treatment dynamics. These visualizations would enhance the manuscript's ability to communicate treatment efficacy effectively.
Reference:
Yau T, et al. Nivolumab Plus Cabozantinib With or Without Ipilimumab for Advanced Hepatocellular Carcinoma: Results From Cohort 6 of the CheckMate 040 Trial. J Clin Oncol. 2023 Mar 20;41(9):1747-1757. doi: 10.1200/JCO.22.00972.
- Subgroup Analysis for Potential Biomarkers
Table 1 provides extensive information on the baseline characteristics of enrolled patients. However, it would be beneficial to explore whether this combination therapy yields differential benefits among subgroups of patients. For instance, stratifying patients by alpha-fetoprotein (AFP) levels (e.g., high vs. low) and generating Kaplan-Meier plots for these subgroups could provide insights into potential biomarkers and predictors of treatment outcomes. This analysis would add significant value to the manuscript by helping to identify which patients are most likely to benefit from the therapy.
Author Response
- Post-Treatment Data Visualization
While the authors provide detailed information on patients' baseline characteristics, there is limited data presented on patients' outcomes during and after treatment. Specifically, only two Kaplan-Meier plots (Figures 1 and 2) are included. To better inform clinicians and researchers, the addition of more detailed post-treatment data is recommended. For example, incorporating a waterfall plot (e.g., Figure 2 from Yau et al.) and a spider plot (e.g., Figure 3 from Yau et al.) would allow for a more comprehensive visualization of individual patient responses and treatment dynamics. These visualizations would enhance the manuscript's ability to communicate treatment efficacy effectively.
Reference:
Yau T, et al. Nivolumab Plus Cabozantinib With or Without Ipilimumab for Advanced Hepatocellular Carcinoma: Results From Cohort 6 of the CheckMate 040 Trial. J Clin Oncol. 2023 Mar 20;41(9):1747-1757. doi: 10.1200/JCO.22.00972.
Response
We sincerely appreciate your valuable comments regarding the additional efficacy data. Following your comment, we added the figures drawing waterfall plot showing best reduction of target lesions and spider plot showing changes of size of target lesions to Figure 3 and Figure 4, respectively. These figures visualizing the efficacy of atezolizumab plus bevacizumab were based on the manuscript you shared with us.
- Subgroup Analysis for Potential Biomarkers
Table 1 provides extensive information on the baseline characteristics of enrolled patients. However, it would be beneficial to explore whether this combination therapy yields differential benefits among subgroups of patients. For instance, stratifying patients by alpha-fetoprotein (AFP) levels (e.g., high vs. low) and generating Kaplan-Meier plots for these subgroups could provide insights into potential biomarkers and predictors of treatment outcomes. This analysis would add significant value to the manuscript by helping to identify which patients are most likely to benefit from the therapy.
Response
We sincerely appreciate your constructive comments regarding the differences of efficacy of atezolizumab plus bevacizumab according to patients’ characteristics. Following your comment, we have performed the subgroup analysis in PFS and OS according to patients’ background, and found the results showed that there were no efficacy differences between groups including AFP levels.We added the subgroup analysis of PFS and OS to Table 3 and Table 4, respectively.
Reviewer 3 Report
Comments and Suggestions for Authors
The topic is interesting and cutting-edge. Maybe the paper is too short and it should be improved, commenting more on the state of the art of systemic treatments in HCC (cite the recent SRMA on lenvatinib PMID: 34017396)
Please add the number at risk and remove the confidence intervals from the KM curves
The limited sample size and the retrospective non-comparative design represent major limitations to the study and they should be addressed as such in the Discussion.
Author Response
The topic is interesting and cutting-edge. Maybe the paper is too short and it should be improved, commenting more on the state of the art of systemic treatments in HCC (cite the recent SRMA on lenvatinib PMID: 34017396)
Response
We appreciated your supportive comment. Following your comment, we have added text to the Introduction section, including the content written in the first paragraph of the Discussion section, about the current status of systemic therapy in advanced hepatocellular carcinoma referred to the article PMID: 34017396, the lack of evidence, and how we make the idea for this study referencing the situation.
Please add the number at risk and remove the confidence intervals from the KM curves
Response
Following your comment, we have added the number at risk to Figure 1 and Figure 2, and remove the confidence intervals from the Kaplan-Meier curves.
The limited sample size and the retrospective non-comparative design represent major limitations to the study and they should be addressed as such in the Discussion.
Response
As you pointed out, this study included a small population and non-comparative design although this was calculated from 80% power at α = 0.05 in verifying the predefined hypothesis. Whereas this trial registered the patients prospectively. Following your comment, we have highlighted these strengths and limitations of this study in the Materials and Methods section and Discussion section as follow.
“This open-labeled, single-arm, prospective phase II trial (jRCT1041200068)” in the Materials and Methods section.
“thus, further investigations are needed to confirm the presented findings.” to the Discussion section following “This study has several limitations such as the analysis with non-comparative design including small population from a single institution” in the same paragraph.
Round 2
Reviewer 2 Report
Comments and Suggestions for Authors
The authors have adequately addressed all my previous comments. Therefore, I recommend this manuscript for publication.
Reviewer 3 Report
Comments and Suggestions for Authors
The revised version of the manuscript is OK. Thank you!